

# Isolation, characterization, and cryopreservation of collared peccary skin-derived fibroblast cell lines

Alana Azevedo Borges[1], Gabriela Pereira De Oliveira Lira[1], Lucas Emanuel Nascimento[1], Maria Valéria De Oliveira Santos[1], Moacir Franco De Oliveira[2], Alexandre Rodrigues Silva[3] and Alexsandra Fernandes Pereira[1]

[1] Laboratory of Animal Biotechnology, Universidade Federal Rural do Semi-Árido, Mossoró, Rio Grande do Norte, Brazil
[2] Laboratory of Applied Animal Morphophysiology, Universidade Federal Rural do Semi-Árido, Mossoró, Rio Grande do Norte, Brazil
[3] Laboratory of Animal Germplasm Conservation, Universidade Federal Rural do Semi-Árido, Mossoró, Rio Grande do Norte, Brazil

Corresponding author
Alexsandra Fernandes Pereira,
alexsandra.pereira@ufersa.edu.br

## ABSTRACT

**Background:** Biobanking of cell lines is a promising tool of support for wildlife conservation. In particular, the ability to preserve fibroblast cell lines derived from collared peccaries is of significance as these wild mammals are unique to the Americas and play a large role in maintaining the ecosystem. We identified collared peccary fibroblasts by immunofluorescence and evaluated their morphology, growth and adherence capacity. Further, we monitored the viability and metabolic activity of the fibroblasts to determine the effects of passage number and cryopreservation on establishment of cell lines.

**Methods:** Skin biopsies were collected from the peripheral ear region from five adult animals in captivity. Initially, cells were isolated from fragments and cultured in the Dulbecco's modified Eagle medium supplemented with 10% fetal bovine serum and 2% antibiotic–antimycotic solution under a controlled atmosphere (38.5 °C, 5% $CO_2$). We evaluated the maintenance of primary cells for morphology, adherence capacity of explants, explants in subconfluence, cell growth and absence of contamination. Moreover, we identified the fibroblast cells by immunofluorescence. Additionally, to evaluate the influence of the number of passages (first, third and tenth passage) and cryopreservation on establishment of cell lines, fibroblasts were analysed for the viability, metabolic activity, population doubling time (PDT), levels of reactive oxygen species (ROS), and mitochondrial membrane potential ($\Delta\Psi$m).

**Results:** All explants (20/20) adhered to the dish in 2.4 days ± 0.5 with growth around the explants in 4.6 days ± 0.7, and subconfluence was observed within 7.8 days ± 1.0. Moreover, by morphology and immunocytochemistry analyses, cells were identified as fibroblasts which presented oval nuclei, a fusiform shape and positive vimentin staining. No contamination was observed after culture without antibiotics and antifungals for 30 days. While there was no difference observed for cell viability after the passages (first vs. third: $P = 0.98$; first vs. tenth: $P = 0.76$; third vs. tenth: $P = 0.85$), metabolic activity was found to be reduced in the tenth passage (23.2 ± 12.1%) when compared to that in the first and third passage (100.0 ± 24.4%, $P = 0.006$). Moreover, the cryopreservation did not influence the viability

($P = 0.11$), metabolic activity ($P = 0.77$), or PDT ($P = 0.11$). Nevertheless, a greater $\Delta\Psi m$ ($P = 0.0001$) was observed for the cryopreserved cells ($2.12 \pm 0.14$) when compared to that in the non-cryopreserved cells ($1.00 \pm 0.05$). Additionally, the cryopreserved cells showed greater levels of intracellular ROS after thawing ($1.69 \pm 0.38$ vs. $1.00 \pm 0.22$, $P = 0.04$).

**Conclusions:** This study is the first report on isolation, characterization and cryopreservation of fibroblasts from collared peccaries. We showed that adherent cultures were efficient for obtaining fibroblasts, which can be used as donor cells for nuclei for species cloning and other applications.

# INTRODUCTION

Collared peccaries (*Pecari tajacu* Linnaeus, 1758) are wild mammals found only in the Americas and show a distribution from southern United States to northern Argentina, inhabiting the most diverse environments (*Santos et al., 2009*). Currently, their population is considered to be stable (*Gongora et al., 2011*); however, a significant reduction of their population has been seen in some biomes, such as the Caatinga (*Desbiez et al., 2012*) and the Atlantic forest (*Lazure et al., 2010*). As excellent seed dispersers (*Redford, 1992*), they are very important for the maintenance of our ecosystem, whereas, economically, they have been commercialized for their meat and in leather production (*Santos et al., 2009*). Scientifically, collared peccaries can be used as experimental models for closely related species such as the *Tayassu peccary* and *Catagonus wagneri* that have been listed as "vulnerable" in the IUCN Red List of Threatened Species (*Keuroghlian et al., 2013*; *Altrichter et al., 2015*; *Gongora et al., 2011*).

In this sense, studies related to the conservation of the collared peccary have been intensified, especially aimed at improving the techniques related to the preservation of somatic samples. Using this study, we established a culture condition for explants derived from the skin of adult collared peccaries (*Santos et al., 2016*) and developed a protocol for cryopreservation (*Borges et al., 2017*, *2018a*, *2018b*) and refrigeration of these explants (*Queiroz Neta et al., 2018*). In order to conduct the cloning experiments on this species by a somatic cell nuclear transfer, as well as to produce induced pluripotent cells, it is necessary to establish properly characterized cell lines.

In general, as observed in other mammals (*Guan et al., 2010*; *Kwong et al., 2014*), establishment of an adequate cell line is a prerequisite step for the success of cloning and producing induced pluripotent cells (*Borges & Pereira, 2019*). For these techniques, fibroblasts and epithelial cells derived from the skin have been widely used (*Jyotsana et al., 2016*; *Siengdee et al., 2018*). Initially, epithelial and fibroblast cells were grown simultaneously; nevertheless, fibroblasts can more easily adhere as well as detach by trypsinization as compared to the epithelial cells (*Bai et al., 2012*; *Saadeldin et al., 2019*; *Siengdee et al., 2018*). In these methods, the culture after the second passage has been considered to contain mainly fibroblasts (*Mehrabani et al., 2014*).

Additionally, for the confirmation of a fibroblast line, it is necessary to verify the possible changes that occur in these cells during culture (*Guan et al., 2010*; *Song et al., 2007*) and cryopreservation (*Magalhães et al., 2017*). In general, the number of passages throughout an in vitro study can modify the cellular epigenetic state, affecting the embryonic development after cloning (*Rodriguez-Osorio et al., 2012*; *Trokovic et al., 2015*). *Magalhães et al. (2017)* observed reduced viability and metabolic activity in the cells derived from the skin of the brown brocket deer in the tenth passage. Thus, the establishment of a cell line ensures a complete knowledge of the parameters that confer quality to the nucleus of the donor cell, named the karyoplast (*Guan et al., 2010*). Moreover, identification of damage occurring during cryopreservation is essential for establishment of a cell line. Cryo-variables may affect several cellular processes, including survival, functionality and the cytoskeleton, which may compromise the reprogramming ability of the karyoplasts (*Chatterjee et al., 2017*). Therefore, we aimed to isolate, characterize and cryopreserve the fibroblast cells derived from the skin of the ear of collared peccaries for their future application in cloning strategies by a somatic cell nuclear transfer and production of induced pluripotent cells.

## MATERIALS AND METHODS

### Chemicals and media

The Dulbecco's modified Eagle medium (DMEM), fetal bovine serum (FBS), penicillin, streptomycin and amphotericin solutions were obtained from Gibco-BRL (Carlsbad, CA, USA). Fluorescent probes were purchased from Invitrogen (Carlsbad, CA, USA). Anti-vimentin antibody and goat anti-mouse IgG (Alexa Fluor® 488, Warrington, PA, USA) were purchased from Abcam (Cambridge, USA). The other chemicals were obtained from Sigma-Aldrich (St. Louis, MO, USA). Media were filtered using a 0.22-μm system (Corning, New York, USA) and adjusted to pH of 7.2–7.4.

### Bioethics and animals

This study was approved by the Ethics Committee of Animal Use of the Federal Rural University of Semi-Arid (CEUA/UFERSA, no. 23091.001072/2015-92) and the Chico Mendes Institute for Biodiversity Conservation (no. 48633-2; ICMBio, Brasilia, Brazil). All animals belonged to the Centre of Multiplication of Wild Animals (CEMAS/UFERSA, Mossoró, RN, Brazil, 5°10′S, 37°10′W), registered at the Brazilian Institute of Environment and Renewable Natural Resources (IBAMA) as a scientific breeding site (no. 1478912). The breeder stocks 100 collared peccaries on an average, and for this research four females and one male at ages of 26.8 months ± 2.9 months were used.

### Ear tissue explant collection and primary culture

Peripheral skin (1–2 cm$^2$) was recovered from the ear sections used to identify collared peccaries kept in captivity. After the collection, a trichotomy of the tissue followed by a sterilization with 70% alcohol was performed. Samples were transported to the laboratory in DMEM supplemented with 2% antibiotic–antimycotic solution (10,000 units/mL

penicillin, 10,000 µg/mL streptomycin and 25 µg/mL amphotericin B) at 37 °C within 30 min.

In the laboratory, fragments (9.0 mm$^3$) were washed sequentially under laminar flow in the following media: (1) DMEM supplemented with 10% FBS and 10% antibiotic–antimycotic solution; (2) alcohol; and (3) DMEM plus 10% FBS and 2% antibiotic–antimycotic solution. Then, the samples were fragmented (four fragments per animal) and placed in polystyrene culture dishes treated for cell adhesion with the latter medium for cell culture. The skin was cultured at 38.5 °C under a controlled environment with 5% CO$_2$ and 95% air, according to a method described by *Santos et al. (2016)*.

## Evaluation of the somatic cells in primary cultures and subcultures

During primary culture, the medium was changed every 24 h. For evaluation of the somatic cells, the primary culture was analyzed before reaching confluency and until it reached a confluency of 70–80%. Using an inverted microscope (Nikon TS100, Tokyo, Japan), the cells were evaluated for the following parameters: cell morphology, number of adhered samples, number of samples, evident subconfluency, day of sample adherence, day of subconfluent growth of the samples, and total time to reach 70–80% confluence (*Borges et al., 2017*).

When the cells reached 70–80% subconfluency, they were subcultured and distributed for other analyses. The 70–80% subconfluence was defined as the stage when 70–80% of the culture dishes consisted of somatic cells (*Santos et al., 2016*). Subconfluent cells were washed with PBS then trypsinized with a trypsin/EDTA solution (0.25%/0.2%) for 7 min and centrifuged at 600×*g* for 10 min. The supernatant was removed, the cell pellet was resuspended in culture medium, and the cell suspension was transferred to another dish for subculturing (*Borges et al., 2018b*). The medium was replaced with fresh medium every other day and the cells were monitored daily. With the successful passaging of the cultures, the cells are considered a cell line, following the convention of the Society of In Vitro Biology (*Schaeffer & Terminology Committee Chair Tissue Culture Association, 1990*). The cell line was designated as Ptskf.

Thus, in addition to an evaluation of the maintenance of cells in the primary culture, the subcultured cells were initially evaluated for the confirmation of fibroblasts using morphology and immunofluorescence analyses. Moreover, the possibility of contamination was also evaluated. Subsequently, the influence of the number of passages (first, third and tenth passage) and the metabolic activity of the cells were analyzed by a viability assay using trypan blue and the 3-(4, 5-dimethylthiazolyl-2)-2, 5-diphenyltetrazolium bromide (MTT) assay, respectively. Moreover, the cells were also evaluated for the effects of a slow freezing cryopreservation. Other than the above-mentioned tests, growth dynamics by quantification of the population doubling time (PDT), oxidative stress analysis for quantification of intracellular reactive oxygen species (ROS) levels using the fluorescent probe 2′,7′-dichlorodihydrofluorescein diacetate (H$_2$DCFDA), and assessment of the mitochondrial membrane potential (ΔΨm) using the fluorescent probe MitoTrackerRed$^®$ were performed.

## Morphological characterization of the fibroblasts

Morphological characteristics were observed throughout the in vitro culture under light microscopy for cellular and nuclear shapes and cytoplasmic extensions.

## Vimentin immunofluorescence

For a morphological confirmation, the cells were subjected to an immunocytochemistry protocol based on the method described by *Amoli et al. (2017)*. Briefly, the cells were fixed using 4% paraformaldehyde for 10 min at 25 °C, then washed with chilled PBS. Subsequently, cells were incubated with an antigen-retrieval buffer (100 mM Tris, 5% urea, pH 9.5), and then permeabilized for 1 h in 0.4% Triton X-100. Afterwards, the cells were incubated in 0.1% Tween-20 for 1 h to block non-specific binding of the antibodies. Finally, the cells were immuno-stained with mouse anti-vimentin antibody (ab8979, 1:200) for 24 h at 4 °C, and, then incubated with the secondary antibody (goat anti-mouse IgG, Alexa Fluor® 488, Warrington, PA, USA, ab150113, 1:400) for 1 h at 25 °C in the dark. Cells were counter-marked with one µg/mL Hoechst for 1 min and observed under a fluorescence microscope (Olympus BX51TF, Tokyo, Japan).

## Confirmation of the absence of bacterial and fungal contamination

Cells of the third passage were cultured for 30 days in DMEM containing 10% FBS in the absence of an antibiotic–antimycotic solution, at 38.5 °C, 5% $CO_2$ and 95% air. Daily evaluation was performed under light microscopy for the identification of bacterial and fungal contamination.

## Influence of the passage number on the quality of fibroblast lines

Initially, the fibroblast cells were analyzed for the effect of the number of passages (first, third and tenth passage) by a viability assay using trypan blue, according to the method described by *Magalhães et al. (2017)*. We evaluated these three cell passages specifically because both fibroblast and epithelial cells were present in the initial (first) passage, only fibroblasts were visualized at the third passage onwards, and the cells of the tenth passage were used for most of the production of embryonic clones (*Shiga et al., 1999*; *Kubota et al., 2000*). The evaluations were performed in triplicate for each animal for each passage.

Briefly, the cells were stained with 0.4% trypan blue in PBS and counted on a hemocytometer. Subsequently, the cells were also analyzed for a metabolic activity using the MTT assay, according to the method described by *Borges et al. (2018b)*. A concentration of $5.0 \times 10^4$ cells/mL from the first, third and tenth passages was grown in 12-well polystyrene plates treated for cell adhesion. After 5 days, 1.5 mL of the MTT solution (five mg/mL in DMEM) was added and the polystyrene culture dishes treated for cell adhesion were incubated for 3 h. The MTT solution was then removed and 1.0 mL of dimethyl sulfoxide (DMSO) was added for 5 min under slow stirring to solubilize the MTT. After the total dissolution of formazan crystals, samples were analyzed in a spectrophotometer (Shimadzu® UV-mini-1240, Kyoto, Japan) at an absorbance

wavelength of 595 nm. The evaluations were performed in triplicate for each animal for each passage.

## Influence of cryopreservation on the quality of fibroblast lines

To evaluate the effect of cryopreservation on the quality of fibroblast lines, cells of the third passage of the five animals were subjected to slow freezing in the freezing medium (DMEM supplemented with 10% DMSO as a permeating cryoprotectant and 10% FBS and 0.2 M sucrose as non-permeating cryoprotectants). Cells at a concentration of $5.0 \times 10^4$ cells/mL were first exposed to DMSO–FBS solution for 15 min at 4 °C, then sucrose solution was added followed by an additional incubation for 15 min at 4 °C. The cryovials containing 1.0 mL of cells in the freezing medium were cooled in a Mr. Frosty freezing container (Thermo Scientific, Waltham, MA, USA) at a cooling rate of 1 °C/min, and later stored in a freezer at −80 °C, reaching −70 °C overnight before being transferred into liquid nitrogen (*León-Quinto et al., 2014*).

For thawing, the cryovials were exposed for 1 min at 25 °C and immersed in a water bath at 37 °C for 3–4 min. Then, the cell contents were removed from the cryovials and washed to remove the cryoprotectants. Initially, the first wash was performed with DMEM and 10% FBS containing 0.2 M sucrose at 4 °C for 15 min and centrifuged. Subsequently, the second wash was performed using only DMEM and 10% FBS, maintained at 25 °C for 15 min, centrifuged, and the cells were recovered for the evaluations as per a method described previously (*Santos et al., 2016*).

After thawing, the non-cryopreserved and the cryopreserved cells were evaluated for growth dynamics by quantification of PDT. The evaluations were performed in triplicate for each animal. Briefly, the growth kinetics was studied for nine days using $3.0 \times 10^4$ cells/mL, and cells were counted daily to determine the number of growing cells. Data on the cell growth and density were monitored and recorded, mean values of which were used to plot a growth curve and calculate PDT (*Roth, 2006*) using the following formula:

PDT = $T \ln 2 / \ln (X_e/X_b)$ where PDT is the time of the culture (in hours), $T$ is the incubation time, $X_b$ is the number of cells at the beginning of the time incubation, $X_e$ is the number of cells at the end of the incubation time, and ln is the Napierian logarithm.

Moreover, for evaluation of an oxidative stress by quantification of the intracellular ROS levels, cells were stained with the fluorescent probe $H_2DCFDA$, according to a method described by *Santos et al. (2019)*. Thawed cells were washed with PBS and placed into polystyrene culture dishes treated for cell adhesion containing 500 μL of 5 μM $H_2DCFDA$. The cells obtained after a 70% confluency were incubated at 38.5 °C in 5% $CO_2$ for 30 min. Stained cells were washed with PBS, placed on glass slides, photographed under a fluorescence microscope (Olympus BX51TF, Tokyo, Japan), and fluorescence signal intensity (pixels) was quantified. Ten images (two/animal) obtained were evaluated using the ImageJ software (version 1.49v, Java 1.8.0_201, Wayen Rasband, U.S. National Institutes of Health, Bethesda, MD, USA; website: http://rsb.info.nih.gov/ij/download. html). The background signal intensity was subtracted from the values obtained for the treated samples. Measured mean value of the micrograph for the non-cryopreserved cells

was taken as a calibrator. Relative expression levels (arbitrary fluorescence units) were generated by dividing the measured value of each micrograph for the cryopreserved cells by the mean of the calibrator.

Finally, for the assessment of ΔΨm, cells were stained using 500 nM of the fluorescent probe MitoTracker Red® (CMXRos), according to a method described by *Santos et al. (2019)*. The procedure, incubation, and evaluation of the ten images (two/animal) were performed as described for the quantification of ROS.

### Statistical analysis

All data have been expressed as the mean ± standard error (one animal/one repetition) and were analyzed using the StatView 5.0 software (Graph-Pad Software Incorporation, La Jolla, CA, USA). Normality of all results was verified by the Shapiro–Wilk test and homoscedasticity was verified by the Levene's test. ROS levels, ΔΨm, viability, and metabolic activity were altered with arcsine and analysed by variance analysis (ANOVA) followed by the Tukey's test. PDT was compared with ANOVA followed by the unpaired *t*-test. Statistical significance was set at $P < 0.05$.

## RESULTS

### Evaluation of the somatic cells in the primary cultures and subcultures

The total culture time was 95 days with an evaluation of cells until the tenth passage. The adhesion of the fragments (Fig. 1A), detachment of cells (Figs. 1B and 1C), and proliferative capacity were observed in all the explants until reaching a confluence (and later, a subconfluence) around the adhered fragments (Figs. 1D–1F; Table 1). All explants had adhesion ability and reached subconfluence. Number of days for each explant to reach a 100% tissue adherence (2.4 days ± 0.5 days), to grow around the explants (4.6 days ± 0.7 day), and to reach subconfluence (7.8 days ± 1.0 day), were different.

### Morphological characterization of the fibroblasts

In cultures, monolayers of cells with a fibroblast-like morphology were observed (Fig. 1E). The cells had an oval nuclei and extensions with a fusiform shape, showing rapid growth that replaced the epithelial cells.

### Vimentin immunofluorescence

Morphology of the fibroblast-like cells in the initial culture was observed by light microscopy, which was further confirmed for the cell type identification as vimentin-labeled fibroblasts under fluorescence microscopy (Figs. 2A–2F). Cells exhibited a high expression of vimentin that marked the cytoplasm completely, and the spindle-like shape and ovoid nucleus was highlighted by the Hoechst labeling. Therefore, the identification of a fibroblast cell was evident.

### Confirmation of the absence of bacterial and fungal contamination

No sign of contamination (turbidity, colony, or hyphal growth) was observed for 30 days in the culture without antibiotics and antifungals. The culture medium did not show any change in the appearance when observed under a light microscope. We did not observe

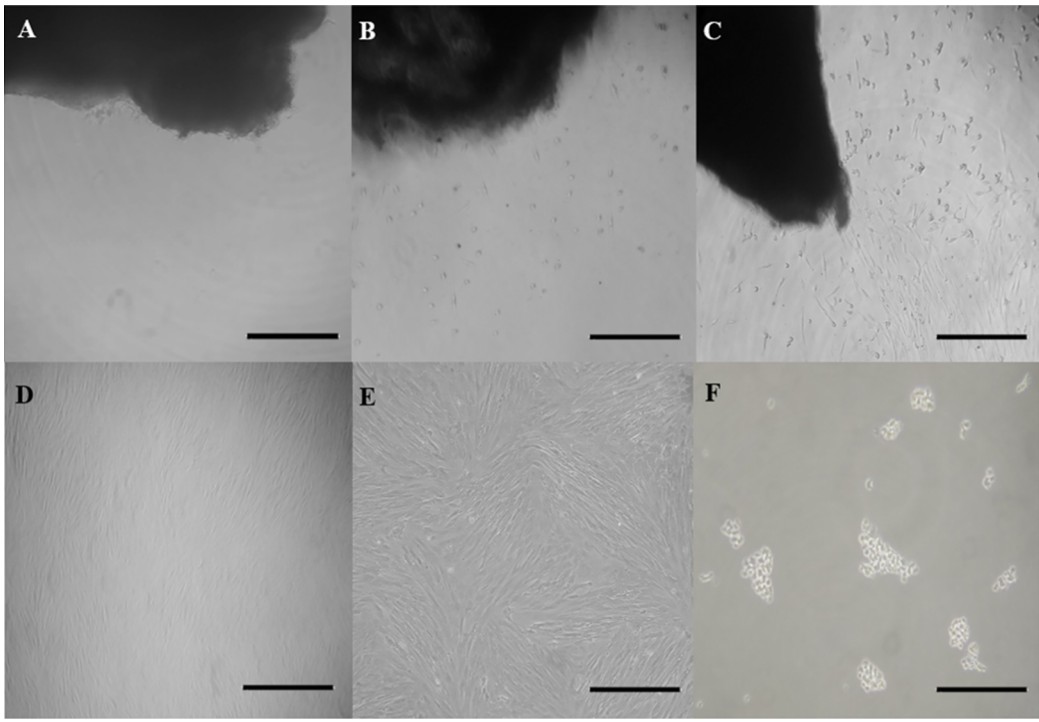

**Figure 1 Outgrowth of fibroblast cells from ear skin samples of collared peccaries.** Exhibit a skin explant cultured in (A) day 1, (B) day 3 and (C) day 5 of primary culture and exhibit a fibroblast population cultured in (D) day 15 and (E) day 19 of subculture and (F) exhibit cells after the trypsinization process. Scale bar = 100 μm.

**Table 1 Establishment of primary culture and subcultures of somatic cells derived from collared peccary ear skin.**

| Animal | No. samples | | | No. attached samples | | |
|---|---|---|---|---|---|---|
| | Initial | Attached (%) | Day all attached explants | Grow to subconfluence (%) | Day all cell grow explants | Subconfluence total time (days) |
| F1 | 4 | 100 | 2 | 100 | 4 | 6 |
| F2 | 4 | 100 | 2 | 100 | 4 | 5 |
| F3 | 4 | 100 | 1 | 100 | 3 | 9 |
| F4 | 4 | 100 | 3 | 100 | 5 | 10 |
| M1 | 4 | 100 | 4 | 100 | 7 | 9 |
| Mean ± S.E. | 20 | 100 | 2.4 ± 0.5 | 100 | 4.6 ± 0.7 | 7.8 ± 1.0 |

turbidity or any specific odor. In addition, there was no change in the biological characteristics of growth and proliferation indicating a complete absence of contamination.

## Influence of the passage number on the quality of fibroblast lines

No significant difference was observed in the cell viability (74.5–84.4%) when evaluated by trypan blue staining after the passages (first vs. third: $P = 0.98$; first vs. tenth: $P = 0.76$; third

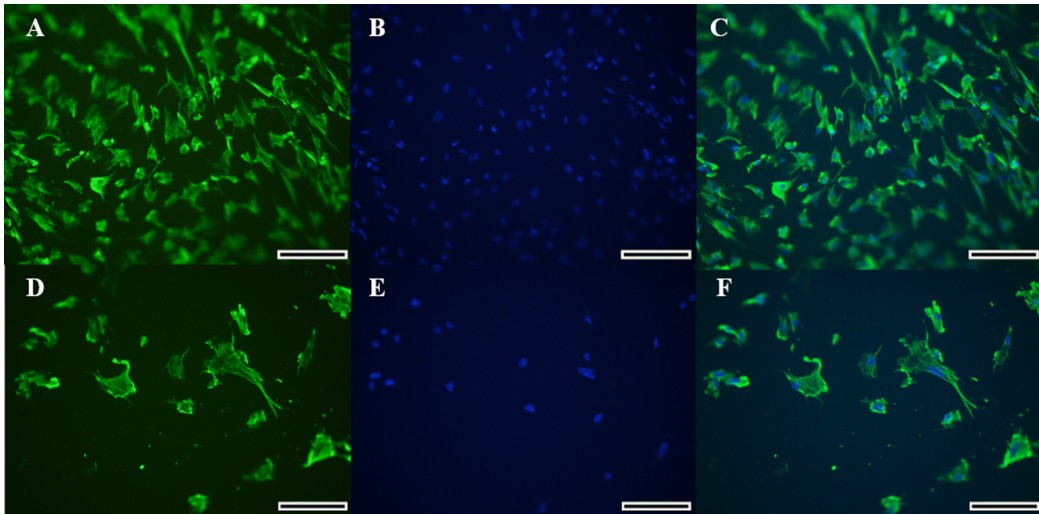

**Figure 2 Immunocytochemical detection of vimetin protein for identification of collared peccary fibroblasts.** (A–D) Cells stained with vimetin antibody. (B–E) Nucleus of cells stained by Hoechst. (C–F) Merged vimetin (green) and Hoechst (blue). A–C (×5), D–F (×10). Scale bar = 10 μm.

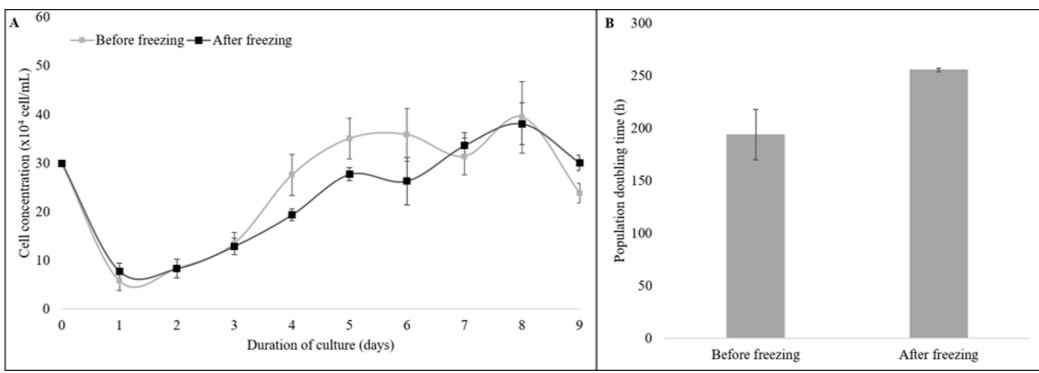

**Figure 3 The growth dynamics of cryopreserved and non-cryopreserved cells derived from collared peccary skin.** (A) Growth curves of cryopreserved and non-cryopreserved cells. (B) Values of population doubling time (PDT) after culture for nine days.

vs. tenth: $P = 0.85$). However, the metabolic activity was reduced in the tenth passage ($23.2 \pm 12.1\%$) as compared to that of the first and third passages ($100.0 \pm 24.4\%$, $P = 0.006$).

### Influence of cryopreservation on the quality of fibroblast lines

Cryopreservation did not affect the viability when evaluated by trypan blue staining ($87.4 \pm 0.3\%$ vs. $74.0 \pm 5.9\%$, $P = 0.11$). Moreover, after two passages of the thawed cells, the viability was $86.4 \pm 3.2\%$. In addition, no difference ($P = 0.77$) was observed for the metabolic activity between the cryopreserved ($85.2 \pm 10.0\%$) and the non-cryopreserved cells ($100.0 \pm 36.4\%$).

Moreover, the cryopreserved and the non-cryopreserved cells were compared for growth dynamics (Fig. 3). The growth curve of both groups showed a typical "S-shaped"

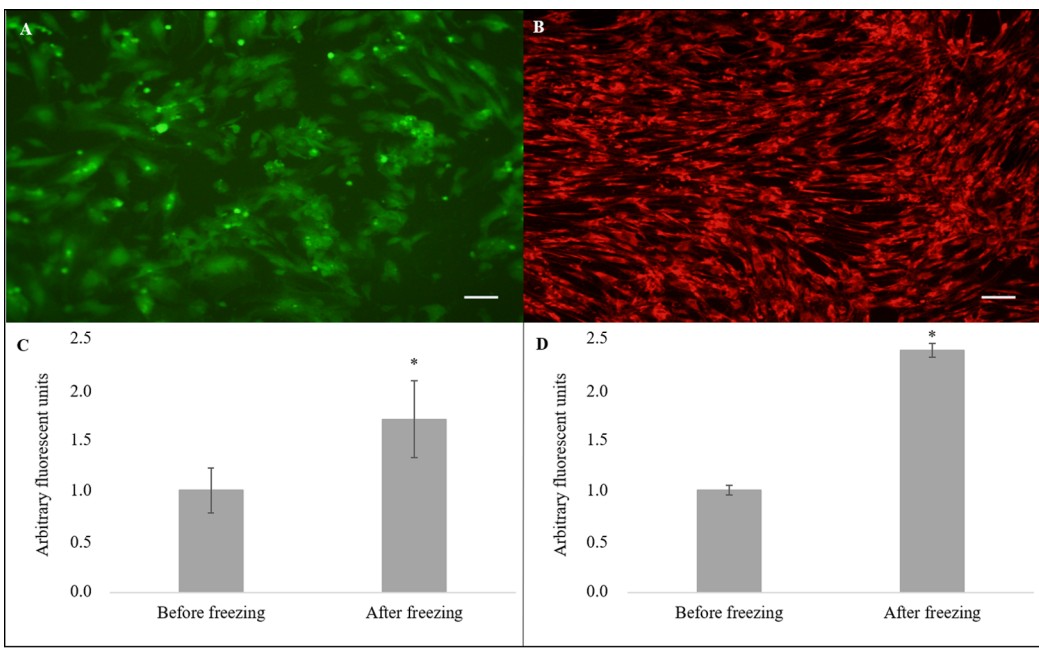

**Figure 4 Evaluation of intracellular reactive oxygen species (ROS) and mitochondrial membrane potential (ΔΨm).** Cell stained using fluorescent probe (A) 2′,7′-dichlorodihydrofluorescein diacetate (H2DCFDA) and (B) MitoTrackerRed® (CMXRos) (×10). Quantification of (C) ROS and (D) ΔΨm levels. Scale bar = 10 μm. (*) Indicate statistical difference ($P < 0.05$).

pattern from the 9-day culture of cells. The latency time was 2 days, followed by an exponential phase until the fourth day, the stationary phase until the 7th day, and the plateau phase from day eight (Fig. 3A). No difference was observed for the PDT values of the cryopreserved and the non-cryopreserved cells ($P = 0.11$, Fig. 3B).

Nevertheless, the cryopreserved cells showed greater levels of intracellular ROS (Fig. 4A) in arbitrary fluorescence units when compared to that of the non-cryopreserved cells ($1.69 \pm 0.38$ vs. $1.00 \pm 0.22$, $P = 0.04$) (Fig. 4C). In addition, an alteration in the ΔΨm (Fig. 4B) in arbitrary fluorescence units ($P = 0.0001$) was observed for the cryopreserved cells ($2.37 \pm 0.07$) when compared to that of the non-cryopreserved cells ($1.00 \pm 0.05$) (Fig. 4D).

## DISCUSSION

In this study, we isolated, characterized, and cryopreserved the fibroblast cells derived from the skin of collared peccaries. Moreover, we established the fibroblast cell lines of these animals with an aim to use these cells in cloning experiments by a somatic cell nuclear transfer in the future. The cell line can be considered as the first constituent of the peccary invitrome and a resource for future studies in many disciplines (*Barioch, 2018*; *Bols et al., 2017*). Thus, the ear tissues of collared peccaries can be isolated and grown into fibroblasts in an adherent culture for establishment of cell lines and development of a cryobank. The development of these somatic cell banks has been increasing in the interest of conserving genetic samples of wild mammals to preserve valuable species, and as

sources for biological research (*León-Quinto et al., 2009*; *Mehrabani et al., 2014*; *Saadeldin et al., 2019*; *Siengdee et al., 2018*).

All explants adhered to the flask surface within 2–4 days, with cellular growth around the explant within 3 days, and demonstrating confluency within 5–10 days after a culture initiation. These characteristics of explants during in vitro culture were similar to the explants derived from other domestic and wild mammals. In studies using tissues from horses, the migration of fibroblast and epithelial-like cells from explants have been observed after 5–7 days of an in vitro culture (*Amoli et al., 2017*). In the case of goat-derived tissues, the explants reportedly adhered to the flasks within 5–7 days and the cells became confluent within 3–5 days post adhesion (*Bai et al., 2012*). In the Iranian Sistani cattle-derived tissues, the explants adhered to the culture flasks within 7–14 days and were observed to allow the growth of fibroblast-like cells from the margins of explants (*Gorji et al., 2017*).

For the Luxi cattle-derived tissues, fibroblast-like or epithelial-like cells could be seen migrating from the tissues within 5–12 days post adhesion (*Liu et al., 2008*). In the tissues derived from wild camels, fibroblast-like or epithelial-like cells could be seen migrating from the sides of explants within 8–10 days post adhesion (*Sharma et al., 2018*). In tissues derived from the domestic porcine, a species phylogenetically close to the collared peccaries, all the explants adhered within 3–8 days (*Silvestre, Sánchez & Gómez, 2004*). The similarity among these data can be related to the culture medium because in a majority of these studies, DMEM containing FBS, antibiotic and antimycotic solution was used (*Magalhães et al., 2017*; *Saadeldin et al., 2019*; *Siengdee et al., 2018*). Since primary culture needs to mimic the in vivo environment of the cells (*Guo et al., 2018*), we observed previously (*Santos et al., 2016*) that the medium for growth of somatic cells derived from collared peccaries was DMEM with 10% FBS and 2% antibiotic–antimycotic solution.

We showed that ear explant cultures obtained from the tissues of the collared peccaries were efficient for cell recovery allowing a culture for 95 days and up to 10 passages. Moreover, we confirmed that the ear skin was a common source where fibroblasts cells could be separated and eventually used as karyoplasts for cloning purposes (*Luo et al., 2014*). These cells were identified by vimentin, an intermediate filament that indicates the mesenchymal origin of endothelial and fibroblast cells (*Yajing et al., 2018*). Initially, during in vitro culture, epithelial and fibroblastic cells grew simultaneously. However, fibroblasts can be trypsinized more rapidly and adhere more easily as compared to the epithelial cells (*Bai et al., 2012*; *Saadeldin et al., 2019*). Therefore, in this work, cells from the third passage were confirmed as fibroblasts by morphology and immunofluorescence analyses.

The clear medium observed during the 30-day assay demonstrated the ability to allow the growth of the culture without any biological contamination. The propagation of fungi causes turbidity by accumulation of their metabolites. In addition, the colonies can be seen under a light microscope, or sometimes with the naked eye (*Li et al., 2009*). Bacterial contamination can also be identified by the naked eye as turbidity. One simple way to avoid this contamination is to filter the culture medium (*Bai et al., 2012*). Sources of

contamination may include, but are not limited to, the equipment, air, culture medium, serum and explant. Therefore, microbial contaminations are quite frequent in cell culture (*Bai et al., 2012*). Thus, the use of antibiotic and antimycotic combined with careful handling, is essential to ensure the absence of contamination.

After two passages, it was possible to separate fibroblast cells from other primary cells because different cell types exhibit different cellular behavior upon trypsinization. The fibroblast cells detach quicker than epithelial cells (*Gorji et al., 2017*). However, fibroblasts detach in response to trypsin more rapidly than epithelial cells and adhere more quickly (*Bai et al., 2012*). In wild camels, the initial lag phase of 48 h representing the adaptation of fibroblasts and recovery from a protease damage is followed by the exponential phase (*Sharma et al., 2018*). In collared peccaries, the replication of cells begins to slow down after 7 days because of contact inhibition, which, in wild camels, has been observed after 6 days (*Sharma et al., 2018*). No difference in the cell viability was observed among the first, third and the tenth passage, corroborating with the studies that used cells from these passages for production of competent cloned embryos (*Shiga et al., 1999*; *Kubota et al., 2000*).

In contrast, through the metabolic activity test evaluated by the formation of formazan crystals, a significant reduction in the metabolic activity at the tenth passage was observed, indicating a reduced cellular functionality. Similar behavior was observed in cells from the brown brocket deer in which the metabolic activity measured by the MTT assay showed significantly lower values in the tenth passage than the values in the fourth passage (*Magalhães et al., 2017*). Therefore, the number of passages can reduce the metabolic activity rate and cell proliferation, thereby conserving cells of the early passages (*Li et al., 2009*). After several passages, genetic characteristics of the cells can be modified by culture conditions; hence, a minimum number of passages have been recommended to conserve the cellular characteristics (*Mehrabani et al., 2014*). Owing to this reason, the cells were cryopreserved in the third passage for the conservation of the somatic germplasm of collared peccaries.

The cell survival rate after thawing is the most commonly used criteria to evaluate the success of a cryopreservation (*Chatterjee et al., 2017*). The cellular viability and the functional metabolic activity of the cells were maintained after thawing the fibroblasts isolated from the collared peccaries. This factor demonstrates that optimal in vitro culture conditions significantly influence the recovery from cellular damages caused by the freezing process (*Gorji et al., 2017*). As for the growth curve, cryopreserved cells presented a very similar profile to that of the non-cryopreserved cells, showing their normal proliferation capacity regardless of the cryopreservation process. The establishment of somatic cell banks using cryopreservation technology is an easy and effective approach towards storing the genetic information of diverse species (*Li et al., 2009*). However, the cells should be handled with the utmost care during cryopreservation to maintain a high-quality cell bank in the long term (*Mehrabani et al., 2014*).

Moreover, epigenetic alterations, such as DNA fragmentation, free radical accumulation, ionic imbalances, apoptosis, biochemical alterations, DNA methylation and

histone modification can be a result of the cryopreservation (*Chatterjee et al., 2017*). These after-effects of cryopreservation may have caused mitochondrial structural abnormalities, thereby promoting an increased ROS production and $H_2O_2$ content, increased lipid peroxidation and increased expression of autophagic proteins harbored by the cells (*Mata et al., 2012*). A failure in the mitochondrial membrane potential is a hallmark of apoptosis, leading to the collapse of the organelle and release of cytochrome-C into the cytoplasm and ultimately activation of the apoptotic cascade (*Magalhães et al., 2012*). Moreover, a high $\Delta\Psi$m mitochondrial respiratory chain becomes a significant ROS producer (*Korshunov, Skulachev & Starkov, 1997*). Therefore, a higher $\Delta\Psi$m in cryopreserved cells can be linked mainly to an increase in the oxidative stress.

Finally, parameters like cryovariables, including cooling and thawing rates, type and concentration of the cryoprotectant, cell type and shape and nucleation temperature may affect the success of cryopreservation (*Chatterjee et al., 2017*). This suggests that the optimization of related cryopreservation methods for the collared peccary fibroblasts to minimize an altered $\Delta\Psi$m and increased levels of intracellular ROS production is essential.

## CONCLUSIONS

To our knowledge, this study is the very first report on a successful isolation, characterization and cryopreservation of fibroblast lines derived from adult collared peccaries (Ptskf). We showed that the adherent culture was efficient for obtaining fibroblasts, which can be used as donor cells for nuclei for cloning of this species. Moreover, it was possible to maintain the viability of the cells until the tenth passage. In addition, cryopreservation did not affect the viability, metabolic activity and proliferative activity of the fibroblasts after slow freezing. However, cryopreservation altered the ROS levels and $\Delta\Psi$m, indicating necessary optimization of the cryopreservation protocol. Lastly, the establishment of fibroblast cell lines derived from collared peccaries may be a source of experimental models for many biological studies such as nuclear reprogramming and animal cloning.

## ACKNOWLEDGEMENTS

The authors thank the Centre for Wild Animals Multiplication (CEMAS/UFERSA) for providing the animals, and the Laboratory Biochemistry and Molecular Biology (BIOMOL/UERN) for technical assistance.

### Funding

This study was supported by the Brazilian Council of Scientific Development (CNPq) and Coordenação de Aperfeiçoamento de Pessoal de Nível Superior— Brasil (CAPES, Financial Code 001). The funders had no role in study design, data collection and analysis, decision to publish, or preparation of the manuscript.

## Grant Disclosures

The following grant information was disclosed by the authors:

Brazilian Council of Scientific Development (CNPq).

Coordenação de Aperfeiçoamento de Pessoal de Nível Superior – Brasil (CAPES): 001.

## Competing Interests

The authors declare that they have no competing interests.

## Author Contributions

- Alana Azevedo Borges conceived and designed the experiments, performed the experiments, analyzed the data, prepared figures and/or tables, and approved the final draft.
- Gabriela Pereira de Oliveira Lira performed the experiments, prepared figures and/or tables, and approved the final draft.
- Lucas Emanuel Nascimento performed the experiments, prepared figures and/or tables, and approved the final draft.
- Maria Valéria de Oliveira Santos performed the experiments, prepared figures and/or tables, and approved the final draft.
- Moacir Franco de Oliveira conceived and designed the experiments, analyzed the data, authored or reviewed drafts of the paper, and approved the final draft.
- Alexandre Rodrigues Silva conceived and designed the experiments, analyzed the data, authored or reviewed drafts of the paper, and approved the final draft.
- Alexsandra Fernandes Pereira conceived and designed the experiments, performed the experiments, analyzed the data, prepared figures and/or tables, authored or reviewed drafts of the paper, and approved the final draft.

## Animal Ethics

The following information was supplied relating to ethical approvals (i.e., approving body and any reference numbers):

This study was approved by the Ethics Committee of Animal Use of the Federal Rural University of Semi-Arid (CEUA/UFERSA, no. 23091.001072/2015-92) and the Chico Mendes Institute for Biodiversity Conservation (ICMBio, no. 48633-2).

## Data Availability

The raw data are available in a Supplemental File.

## Supplemental Information

Supplemental information for this article can be found online at http://dx.doi.org/10.7717/peerj.9136#supplemental-information.

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
