# Peer review of "Isolation, characterization, and cryopreservation of collared peccary skin-derived fibroblast cell lines"

_PeerJ, doi:10.7717/peerj.9136_

## Round 0.1 · original submission · Major Revisions

There are some methodological and terminological ambiguities that were highlighted by reviewers 1, 3 and 4. I consider the general manuscript style appropriate and the manuscript to be generally well written, but the manuscript will be subjected to editorial review prior to possible publication.

·

Basic reporting

The work of Borges et al. is a continuation of the previous works of the same group on collared peccary. They reported the isolation, characterization, and cryopreservation of fibroblasts from the collared peccary.

Experimental design

There is incomplete information and data presentation that should be revised in the current version of the manuscript.

Validity of the findings

The presented data requires more clarification.

Additional comments

There is incomplete information and data presentation that should be revised in the current version of the manuscript.

L143: Provide information about the trypsinization process (Concentration of trypsin and duration for cell separation) that was used for fibroblast isolation from other skin-derived cells, and also that was used for subsequent passaging.

L150 & L184: Provide figures for the cultured cells in bright-field or normal light microscope. Fibroblasts are mostly spindle morphology, however, in the provided figure of vimentin the cells look irregular shape. Also, provide if you have ever observed epithelial cells, keratinocytes and other cells with the primary cultured explants.

L162: I suggest separating this sub-heading into 2 parts: (1) The morphological characterization of fibroblast, (2) Vimentin immunofluorescence.

Figure 1: I think there are some missed parts of Fig. 1. Describe the abbreviations EG, SUC, and DMSO in the figure caption. What are A, B, C, D, and E? What is meant by scale bar?!

Figure 2: Revise the caption; this is a “merged” image, not “double-stained”.

Figure 3: Why there is a drop in cell viability from 30 to less than 10 X 10^4 in both cells? Is trypsin used appropriately? In L287: latency does not mean a drop in the cell viability if the cells were perfectly dissociated. Moreover, Population Doubling Time is very long, it is more than 8 days! It is very long when compared to your previous report, which was around 2-3 days. https://doi.org/10.1007/s11626-018-0270-6

Table 1: Delete the data of “total number or sum” for avoiding misunderstanding.

L48, and throughout the manuscript: you provide the actual P-value, therefore, replace “>” with “=”.

L50: Indicate the P-value.

For paragraph 84-90: support this statement with recent trials for isolation of skin-derived fibroblasts from other mammals and the usage of trypsin for discrimination between different cells, and cite these two references: (doi: 10.7717/peerj.4302 & doi: 10.3390/ani9060378). You can use the results of these two references for the comparative skin cell culture that you have described in L309-329.

L292: Provide images for results of ROS fluorescence and mitochondrial membrane potential.

The title: please correct “skin-derived fibroblast cell lines”.

·

Basic reporting

In the manuscript, the authors reported that they have established fibroblast somatic cell lines from skin biopsies of collared peccaries. They reported that the cryopreservation process does not affect the viability, metabolic activity, and proliferative activity, except ROS and ΔΨm. The established somatic cells are prerequisites for somatic cell nuclear transfer experiments and other biological experiments. In my view, established somatic cell lines can be used for animal cloning experiments in the future.

Experimental design

This study aimed to establish somatic cell lines of collared peccaries. The multiple experiments were made to prove that methods used to establish somatic cells are efficient, and all culture conditions supported the efficient culture and maintenance of established cells.

Validity of the findings

the results are well stated, and all findings supported the original research question.

Additional comments

In the title, cell lines instead of cell line, since, five cell lines established.

Reviewer 3 ·

Basic reporting

1. Manuscript was poorly written. The writing of the manuscript should be reviewed.
2. The introduction does not justify the relevance of the objectives set at work.
• The authors present a work based on the isolation of fibroblasts from adult specimens of Collared Peccary (pecari tajacu) to generate a biobank for conservation purposes.
The International Union for Conservation of Nature (IUCN) considers stable the pecari tajacu population and only establishes as objectives to investigate (https://www.iucnredlist.org/species/41777/10562361#conservation-actions):
• Population size, distribution & trends.
• Harvest, use & livelihoods
• Population trends.
• The authors justify the objectives of the work by being aimed at the conservation of other species phylogenetically (White-lipped Peccary-tayassu peccary and Chacoan Peccary-catagonus wagneri). In the discussion of the work, they justify that the isolation of fibroblasts will allow their use in cloning by SCNT. But the use of SCNT for biological conservation requires that the donor cell belongs to the threatened species.
Moreover, until now SCNT has not had the potential to be considered as a true option for biological conservation efforts.
c) For the establishment of a somatic cell bank of an endangered species, the isolation of pluripotent or multipotent cells such as mesenchymal stem cells (Liu et al 2013. Stem cell and development) is of greater relevance.

3. The figure legend 1 describes containing 5 figures: A, B, C, D and E. On the other hand, the image corresponding to figure 1 does not correspond to the figure legend and consists of only 1 figure.

Experimental design

1. The sample size is statistically insufficient and does not present a correct sex ratio nor show adequate age distribution. The authors have only used 5 specimens of the species, four females and one male and all adults.
2. The different experiments do not show n or the number of replicates performed. Cell viability by trypan blue, metabolic activity, ROS and population doubling time analyses do not show n and number of replicates.

Validity of the findings

The experiments have not been done with sufficient scientific rigor, nor are they explained with sufficient clarity.
1. How many times did the authors clean the tissue samples and what solution did they use? How did the authors clean the tissue samples? Centrifugation? Speed?
2. What is the subculture ratio?
3. The authors do not mention the surface of the culture plates used.
4. What are penicillin, streptomycin amphotericin, and Trypsin concentration?
5. What are the cellular passage and thawing protocols used?
6. For cell viability, how many cells were determined from a field?
7. How many replicates were done for each technique?
8. For detection of primary cultures contamination, the authors do not use any specific technique for bacterial, fungi (Doyle et al 1990) mycoplasma (Masover and Becker 1998; Freshney's method 2000) or virus detection, using positive controls.
9. Authors do not perform a karyotype analysis of the cell cultures.
10. Figure 3: cell proliferation after the 8th day was reduced. The reason is the cells began to enter the plateau phase because of contact inhibition?

Reviewer 4 ·

Basic reporting

no comment

Experimental design

no comment

Validity of the findings

no comment

Additional comments

Line 147 Could you the authors consider adding the following sentences?

With the successful passaging of the cultures, the cells are considering a cell line, following the convention of the Society of In Vitro Biology (Schaeffer, 1990). The cell line has been designated Ptskf. (Ptskf is just a suggestion).

Lines 302 Could you the authors consider adding the following sentences?
The cell line can be considered the first constituent of the peccary invitrome and a resource for future studies in many disciplines (Barioch, 2018; Bols et al., 2017).

Lines 353-354 Could the authors recast the sentence as follows? However, fibroblasts detach in response to trypsin more rapidly than epithelial cells and adhere more quickly (Bai et al., 2012).

Could the authors please consider adding the following references?
Barioch A (2018). The Cellosaurus: a cell knowledge resource. J Biomol Tech 29: 25-38.

Bols NC, Pham PH, Dayeh VR & LEJ Lee (2017). Invitromatics, invitrome, and invitroomics: Introduction of three new terms for in vitro biology and illustration of their use with the cell lines from rainbow trout. In Vitro Cellular & Developmental Biology-Animal 53: 383-405.

Schaeffer WI (1990) Terminology associated with cell, tissue and organ culture, molecular biology and molecular genetics. In Vitro Cell Dev Biol 26: 97-101.

---

## Round 0.2 · accepted · Accept

Thank you for the careful address of the reviewers' concerns. There will be suggestions for grammatical corrections forthcoming.

·

Basic reporting

The authors have addressed my comments and suggestions.

Experimental design

The methods have been improved accordingly.

Validity of the findings

Results are now supported by the images.

Additional comments

The authors have addressed my comments and suggestions. The manuscript has been improved accordingly.

Reviewer 3 ·

Basic reporting

The authors have adequately responded to the proposed corrections and suggestions, have resolved the errata and have successfully discussed some problematic aspects of the article.

Experimental design

ok

Validity of the findings

ok